# The Antimicrobial Peptide AMP-IBP5 Suppresses Dermatitis-like Lesions in a Mouse Model of Atopic Dermatitis through the Low-Density Lipoprotein Receptor-Related Protein-1 Receptor

**DOI:** 10.3390/ijms24065200

**Published:** 2023-03-08

**Authors:** Hai Le Thanh Nguyen, Ge Peng, Juan Valentin Trujillo-Paez, Hainan Yue, Risa Ikutama, Miho Takahashi, Yoshie Umehara, Ko Okumura, Hideoki Ogawa, Shigaku Ikeda, François Niyonsaba

**Affiliations:** 1Atopy (Allergy) Research Center, Juntendo University Graduate School of Medicine, Bunkyo-ku, Tokyo 113-8421, Japan; 2Department of Dermatology and Allergology, Juntendo University Graduate School of Medicine, Bunkyo-ku, Tokyo 113-8421, Japan; 3Faculty of International Liberal Arts, Juntendo University, Bunkyo-ku, Tokyo 113-8421, Japan

**Keywords:** antimicrobial peptide, AMP-IBP5, atopic dermatitis, barrier function, skin barrier

## Abstract

The antimicrobial peptide derived from insulin-like growth factor-binding protein 5 (AMP-IBP5) exhibits antimicrobial activities and immunomodulatory functions in keratinocytes and fibroblasts. However, its role in regulating skin barrier function remains unclear. Here, we investigated the effects of AMP-IBP5 on the skin barrier and its role in the pathogenesis of atopic dermatitis (AD). 2,4-Dinitrochlorobenzene was used to induce AD-like skin inflammation. Transepithelial electrical resistance and permeability assays were used to investigate tight junction (TJ) barrier function in normal human epidermal keratinocytes and mice. AMP-IBP5 increased the expression of TJ-related proteins and their distribution along the intercellular borders. AMP-IBP5 also improved TJ barrier function through activation of the atypical protein kinase C and Rac1 pathways. In AD mice, AMP-IBP5 ameliorated dermatitis-like symptoms restored the expression of TJ-related proteins, suppressed the expression of inflammatory and pruritic cytokines, and improved skin barrier function. Interestingly, the ability of AMP-IBP5 to alleviate inflammation and improve skin barrier function in AD mice was abolished in mice treated with an antagonist of the low-density lipoprotein receptor-related protein-1 (LRP1) receptor. Collectively, these findings indicate that AMP-IBP5 may ameliorate AD-like inflammation and enhance skin barrier function through LRP1, suggesting a possible role for AMP-IBP5 in the treatment of AD.

## 1. Introduction

Atopic dermatitis (AD) is a common chronic inflammatory skin disease with a complex and multifactorial etiology involving genetic, immunological, and environmental factors that elicit immune dysfunction and skin barrier disruption [1,2]. Cumulative evidence has shown that, similar to immune dysfunction, disruption of the skin barrier plays a central role in the pathogenesis of AD. Impairment of tight junctions (TJs) has consistently been reported to be a key player in AD pathogenesis, as TJs interconnect with other components of the skin barrier [3,4,5]. TJs are intercellular junctions located within the stratum granulosum (SG) and comprise the claudin, occludin, and zonula occludens (ZO) protein families. TJs are thought to shape a paracellular permeability barrier regulating the passage of water, ions, and solutes [6]. De Benedetto et al. reported a significant reduction in the expression of claudin-1, an important TJ-related protein in the epidermal barrier, in subjects with AD, suggesting the deficiency of TJs in AD patients [7]. Claudin-1 deficiency was demonstrated to cause increases in transepidermal water loss (TEWL) levels in mice, accompanied by notable decreases in skin barrier function [8]. In addition, TJ barrier defects were indicated to inhibit the formation of the stratum corneum (SC) by altering polar lipid and profilaggrin processing, suggesting that TJ failure might be correlated with dysfunction of the SC barrier in AD pathogenesis [9,10].

Antimicrobial peptides (AMPs) constitute the first-line of defense in the skin against pathogens. In addition to exhibiting wide-ranging killing activity against pathogenic microorganisms, AMPs also exhibit immunomodulatory activity by inducing cell proliferation, migration, and differentiation [11]. They regulate cytokine/chemokine production, enhance angiogenesis and wound healing, and maintain skin barrier function [11,12,13,14]. The antimicrobial peptide derived from insulin-like growth factor-binding protein 5 (AMP-IBP5) is a proteolytic cleavage product of insulin-like growth factor-binding protein (IGFBP)-5, which belongs to the IGFBP family. This family consists of six proteins, namely, IGFBP-1 to IGFBP-6 [15,16]. IGFBP-5 is expressed in various cells, including keratinocytes and fibroblasts [16,17]. AMP-IBP5 has been found to accelerate diabetic wound healing through its protective effects against glucotoxicity and promoting effects on angiogenesis. These functions reveal a promising role of AMP-IBP5 in the management of chronic wounds [18]. Additionally, AMP-IBP5 was reported to induce the proliferation and migration of both keratinocytes and fibroblasts through the receptor low-density lipoprotein receptor-related protein-1 (LRP1) [19]. LRP1 is a member of the low-density lipoprotein (LDL) receptor family. It comprises a membrane-anchoring 85-kDa subunit and a 515-kDa extracellular chain with four ligand-binding clusters [20,21]. LRP1 is expressed in several organs, including liver, lung, brain, and skin [22]. In the skin, LRP1 expression is detected mainly in the SG of the epidermis and in dermal fibroblasts [23].

Various AMPs, such as LL-37, human β defensins (hBDs), and S100A7, have been demonstrated to be involved in the mechanism regulating TJ barrier function in keratinocytes in vitro [24,25,26,27]. Because skin barrier dysfunction plays a central role in the pathogenesis of AD [28,29,30], AMPs might have therapeutic promise in the management of AD. Here, we aimed to clarify the effect of AMP-IBP5 on skin barrier function in the context of AD and to identify its underlying mechanism in order to determine a possible therapeutic approach for AD.

## 2. Results

### 2.1. AMP-IBP5 Improves TJ Barrier Function

We first investigated whether AMP-IBP5 can affect the expression of TJ-related proteins in human primary keratinocytes. Western blot analysis revealed that AMP-IBP5 markedly upregulated claudin-1, -4, and -7, occludin and ZO-1 expression (Figure 1). Because the intercellular distribution of TJ-related proteins is indispensable for the formation of functional TJs, the effect of AMP-IBP5 on the intercellular localization of TJ-related proteins was examined. AMP-IBP5 induced a broad distribution of claudin-1, -4, and -7, occludin and ZO-1 along intercellular borders (Appendix A). In addition, to examine whether the AMP-IBP5-mediated intercellular localization of TJ-related proteins is indeed associated with TJ barrier function, we assessed the effect of AMP-IBP5 on transepithelial electrical resistance (TER), a sensitive measure of TJ barrier function [31,32]. Upon stimulation of keratinocyte monolayers with AMP-IBP5, the TER values dose- and time-dependently increased and peaked at 48 h, before gradually declining (Appendix A). Collectively, our results indicate that AMP-IBP5 enhances the formation and function of the TJ barrier.

### 2.2. AMP-IBP5 Enhances TJ Barrier Function through Activation of Atypical Protein Kinase C (aPKC) ζ and Ras-Related C3 Botulinum Toxin Substrate 1 (Rac1)

The two isoforms of the aPKC, namely, aPKCζ and aPKCi/λ, have been indicated to be associated with TJ barrier function [33,34]. Furthermore, activation of guanosine triphosphate (GTP)-bound Rac1 (GTP-Rac1) is fundamental for the formation of the TJ barrier and the maturation of epidermal keratinocytes [35]. Additionally, GTP-Rac1 activation is essential for the activation of aPKCζ [36]. Thus, we investigated whether AMP-IBP5 regulates TJ barrier function through activation of aPKC and GTP-Rac1. Treatment of human keratinocytes with AMP-IBP5 markedly increased the phosphorylation of aPKCζ, which peaked at 30 min. Phosphorylation of Rac1 (at Ser71) was observed 1 and 2 h post-stimulation (Figure 2A). Activation of both aPKC and Rac1 was necessary for AMP-IBP5-mediated regulation of the TJ barrier, as demonstrated by the inhibitory effects of GF 109203X, a pan-PKC inhibitor [37], and NSC23766, a specific inhibitor of Rac1 [38]. Indeed, the presence of GF 109203X notably suppressed the protein expression of claudin-1, occludin, and ZO-1 (Appendix A) and impeded the intercellular localization of these TJ-related proteins (Figure 2B, upper panels). Moreover, treatment of keratinocyte monolayers with GF 109203X significantly suppressed the AMP-IBP5-induced increase in the TER (Figure 2C, left panel). Similarly, NSC23766 noticeably suppressed the expression of claudin-1, occludin, and ZO-1 (Appendix A), impeded their intercellular localization (Figure 2B, lower panels), and decreased the TER (Figure 2C, right panel) in AMP-IBP5-treated keratinocytes. These observations suggest that activation of aPKCζ and Rac1 is needed for the AMP-IBP5-induced enhancement of TJ barrier function.

### 2.3. AMP-IBP5 Regulates TJ Barrier Function via the LRP1 Receptor

AMP-IBP5 was reported to induce the proliferation and migration of keratinocytes and fibroblasts through LRP1 [19]; thus, we hypothesized that AMP-IBP5 might regulate TJ barrier function through LRP1. To confirm our hypothesis, we pretreated keratinocytes with LDL receptor-related protein-associated protein 1 (LRPAP), also named RAP, an antagonist of LRP1 [39]. The presence of RAP significantly impeded the intercellular localization of the TJ-related proteins claudin-1, occludin, and ZO-1 (Figure 3A), and decreased the protein expression of these TJ-related proteins (Appendix A). Additionally, pretreatment of keratinocyte monolayers with RAP suppressed the AMP-IBP5-induced increase in the TER (Figure 3B) and the phosphorylation of aPKCζ and Rac1 (Figure 3C). These findings suggest that AMP-IBP5 regulates TJ barrier function through LRP1 and that aPKCζ and Rac1 function downstream of this receptor.

### 2.4. Treatment with AMP-IBP5 Rescues IL-4/IL-13-Driven TJ Barrier Dysfunction

TJ barrier failure, which was indicated to be involved in the pathogenesis of AD, not only disrupts the epidermal barrier but also impacts the immune response [3,4,5,40]. Interestingly, our analysis of microarray data from lesional and non-lesional skin of patients with AD revealed that the gene expression of IGFBP-5, the parent protein of AMP-IBP5, is downregulated in AD skin lesions (Figure 4A). Based on this observation, we examined the expression of IGFBP-5 in a mouse model of 2,4-dinitrochlorobenzene (DNCB)-induced AD-like pathology and found that the mRNA expression of IGFBP-5 was decreased in AD mice compared to normal mice (Figure 4B). This finding suggests that AMP-IBP5 deficiency may play a crucial role in the pathogenesis of AD.

To test this hypothesis, we treated human primary keratinocytes with IL-4 and IL-13 to establish an in vitro model that mimics AD features. IL-4/IL-13 treatment of keratinocytes impaired the spontaneous intercellular localization and inhibited the protein expression of claudin-1, occludin, and ZO-1 (Figure 5A,B) and decreased the TER (Figure 5C). Interestingly, the presence of AMP-IBP5 enhanced the intercellular localization and increased the protein expression of TJ-related proteins (Figure 5A,B) and increased the TER of keratinocyte monolayers pretreated with IL-4/IL-13 (Figure 5C). These results indicate that the addition of AMP-IBP5 might rescue IL-4/IL-13-driven TJ barrier dysfunction.

### 2.5. AMP-IBP5 Ameliorates AD Symptoms in Mice with DNCB-Induced AD-like Pathology

To further verify whether AMP-IBP5 improves skin barrier function in vivo, we established a model of DNCB-induced AD in BALB/c mice. The lesional ear skin of AD mice was treated subcutaneously with AMP-IBP5. AD model mice exhibited significant increases in dermatitis scores, ear thickness, scratching frequency, and TEWL. Although significant differences remained between normal mice and AMP-IBP5-treated AD mice, these features of AD were markedly improved following AMP-IBP5 treatment (Figure 6A–C). Moreover, the expression of claudin-1, the most important TJ-related protein, was significantly decreased in AD mice, and treatment with AMP-IBP5 noticeably restored claudin-1 expression (Figure 6D and Appendix A). Intriguingly, no differences in the alleviation of dermatitis-like symptoms in the AD mouse model, including dermatitis scores, ear thickness, scratching frequency, TEWL (Appendix A), and the expression of claudin-1 (Appendix A), were observed between mice treated with AMP-IBP5 via topical application and those treated via injection.

To confirm whether the AMP-IBP5-mediated enhancement of TJ protein expression is associated with a functional TJ barrier in AD mice, we performed a TJ permeability assay using an NHS-LC-biotin tracer, as described previously [41]. In normal mice with a functional TJ barrier, the tracer does not pass through the outermost layer of the epidermis, whereas in mice with a disrupted TJ barrier the tracer does so readily [8,42]. As shown in Figure 6E, the penetration of the tracer (red) into the epidermis of normal mice was blocked (arrowheads) by the functional TJ barrier, which is represented by claudin-1 (green dots). However, in AD mice, the tracer was able to pass through the epidermis. As expected, after AMP-IBP5 treatment, tracer penetration was again blocked by the TJ barrier, indicating that AMP-IBP5 treatment restored TJ barrier function in AD mice.

Interestingly, while the expression levels of type 2 cytokines IL-4, IL-13, and IL-33 and pruritic cytokines, such as IL-31 and thymic stromal lymphopoietin (TSLP), were increased in AD skin lesions, administration of AMP-IBP5 decreased the expression of type 2 and pruritic cytokines (Figure 7A). Injection of AMP-IBP5 into the skin lesions of AD mice also noticeably reduced the numbers of CD4^+^ T cells (Figure 7B) and mast cells (Figure 7C) and the total serum immunoglobulin (Ig) E level (Figure 7D).

### 2.6. LRP1 Is Required for the AMP-IBP5-Mediated Amelioration of AD

To elucidate the role of LRP1 in the AMP-IBP5-mediated amelioration of AD symptoms, we subcutaneously administered mouse RAP, an antagonist of LRP1, to AD mice to inhibit LRP1. In our model, co-treatment with AMP-IBP5 and RAP did not result in improvements in the dermatitis score, ear thickness, scratching frequency, or TEWL in AD mice (Appendix A and Figure 8A,B). Furthermore, in the presence of RAP, AMP-IBP5 failed to increase the epidermal expression of claudin-1 (Figure 8C) or enhance skin barrier function, as evaluated by a permeability assay (Figure 8D), in AD mice.

Additionally, inhibition of LRP1 in AD mice blocked the effects of AMP-IBP5 on reducing the expression of type 2 and pruritic cytokines and the total IgE serum level (Figure 9A). AMP-IBP5 also failed to reduce the numbers of CD4^+^ T cells (Figure 9B) and mast cells (Figure 9C) in the presence of RAP in AD mice. Collectively, these results indicate that AMP-IBP5 ameliorates AD through LRP1.

## 3. Discussion

Skin barrier dysfunction contributes significantly to the pathogenesis of AD. The TJ barrier consistently plays a crucial role, as it interconnects with other components of the skin barrier [8,40,43]. Interestingly, AMPs such as hBDs, LL-37, and S100A7 have been reported to improve the TJ barrier in vitro in keratinocyte monolayers [24,25,26,27]. Although accumulated evidence implies a promising role for AMPs in the treatment of AD [11,40,44], the precise role of AMPs in the immunopathogenesis of AD remains elusive. In this study, our microarray analysis revealed that the gene expression of IGFBP-5, the parent protein of AMP-IBP5, is lower in skin lesions of patients with AD than in non-lesional skin, suggesting that AMP-IBP5 might be involved in AD pathogenesis. Moreover, we showed that AMP-IBP5 increased the expression and enhanced the intercellular distribution of various TJ-related proteins and improved skin barrier function in both in vitro and in vivo AD models. The crucial role of the SC in the permeability barrier has been extensively demonstrated. However, the importance of the TJ barrier has been a recent research focus, particularly in skin diseases characterized by skin barrier defects, such as AD and psoriasis. In fact, knockout of claudin-1 in mice leads to an increase in TEWL and mortality with marked functional impairment of the skin barrier, indicating the importance of the epidermal barrier and TJ proteins [8]. Claudin-1 expression was found to be significantly decreased in skin lesions from patients with AD compared with healthy skin from nonatopic individuals [45]. Moreover, Tokumasu et al. indicated that claudin-1 orchestrates the features of AD and has a potential role in the pathogenesis, severity, and natural course of AD [46]. Herein, we demonstrated that AMP-IBP5 restores both claudin-1 expression and barrier function in AD mice and ameliorates AD symptoms in these mice, further indicating the crucial role of the TJ barrier in AD. There was no significant difference in alleviation of dermatitis-like symptoms between subcutaneous injection and topical application of AMP-IBP5. According to the 500 Dalton rule for the skin penetration of chemical compounds [47], AMP-IBP5 with a molecular weight of 2655 Dalton hardly penetrates the skin. We speculated that AMP-IBP5 succeeded in penetrating the skin because it was dissolved in acetic acid, which is known as a skin penetration enhancer [48,49]. Moreover, skin barrier impairment in AD may also facilitate the penetration of AMP-IBP5 into the skin. Further studies are required to clarify the topical effect of AMP-IBP5 in AD pathogenesis.

LRP1 plays an important role not only in the regulation of lipoprotein metabolism but also in numerous aspects of cell signaling and function [22,50]. In the skin, LRP1 is expressed in various cell types, including keratinocytes, fibroblasts, dendritic cells, and endothelial cells, and is involved in the regulation of skin homeostasis [23,51]. Moreover, LRP1 plays an important role in regulating the host immune response to invading pathogens. In fact, LRP1 is involved in the innate recognition of microbial components and the inflammatory response in macrophages [52]. Here, our findings indicated that LRP1 might participate in regulating the TJ barrier in human keratinocytes and in mice with DNCB-induced dermatitis-like lesions, because AMP-IBP5 failed to ameliorate AD in RAP-treated keratinocytes and AD mice. These data reveal a critical role of LRP1 in the barrier-based pathogenesis of AD.

LRP1 functions as a major regulator of the activation of mitogen-activated protein kinases, Rac1, and aPKCζ in several types of cells. For instance, LRP1 participates in the activation of mitogen-activated protein kinase cascades induced by AMP-IBP5, contributing to the proliferation and migration of keratinocytes and fibroblasts [19]. LRP1 was also demonstrated to be involved in Rac1-regulated murine embryonic cell migration [53] and in aPKCζ-regulated chondrocyte differentiation [54]. Therefore, our observation that AMP-IBP5 regulates the TJ barrier in human keratinocytes through activation of aPKCζ and Rac1 is consistent with previous studies. To further clarify the role of LRP1 in skin barrier regulation, we investigated whether AMP-IBP5 administration is effective in AD mice with LRP1 inhibition. AMP-IBP5 administration exhibited no therapeutic effect in AD mice with LRP1 inhibition, confirming that LRP1 is required for the AMP-IBP5-mediated amelioration of AD. To our knowledge, this is the first report about the involvement of LRP1 in the pathogenesis of AD. Further studies are required to investigate the mechanism by which LRP1 is involved in the pathogenesis of AD.

Recent studies have revealed the involvement of AMPs in the pathogenesis of inflammatory skin diseases, in which overproduction of AMPs may exacerbate the inflammatory component of the disease. Indeed, hBDs activate T cells and mast cells to produce IL-4, IL-13, and IL-31, characteristic inflammatory cytokines in AD [55,56]. Activated T cells upregulate the production of Th2 cytokines, including IL-31, interferon-γ, and IL-22, in the presence of LL-37, indicating that LL-37 promotes the inflammatory environment in individuals with T-cell-related skin diseases [57]. hBDs and LL-37 were also demonstrated to promote the production of inflammatory cytokines in mast cells [56,58]. In addition to exhibiting proinflammatory activities, AMPs might also exert anti-inflammatory effects. For instance, LL-37 exerts an antagonistic effect on interferon-γ, tumor necrosis factor-α, IL-4, and IL-12 responses in several cell types [59,60,61]. Intriguingly, we demonstrated that AMP-IBP5 administration suppressed the expression of IL-4, IL-13, IL-31, IL-33, and TSLP in mice with DNCB-induced AD. Moreover, AMP-IBP5 treatment reduced the numbers of CD4^+^ T cells and mast cells and the total serum IgE level, suggesting a possible anti-inflammatory effect of AMP-IBP5 in the AD mouse model.

AD is characterized by chronic eczematous lesions, skin dryness, and intense pruritus [62]. Scratching due to chronic pruritus in AD further exacerbates skin barrier dysfunction [63], causes sleep loss, and critically decreases the quality of life of AD patients [64,65]. Notably, hBDs and LL-37 stimulate mast cells to produce IL-31, which is strongly involved in the regulation of itch sensation in patients with AD [56,66]. Although hBDs and LL-37 have also been shown to be involved in the production of Th2 inflammatory and pruritic cytokines [12,67,68,69,70], they also inhibit the production of pruritic cytokines. Indeed, LL-37 promotes the production of semaphorin 3A, a chemorepulsive factor of the epidermal nerves that is downregulated in AD [71], and suppresses TSLP production in keratinocytes [61]. These functions indicate a possible role of LL-37 in itch inhibition in patients with AD. In this study, AMP-IBP5 suppressed the expression of IL-31 and TSLP and ameliorated pruritus in AD mice, suggesting its potential involvement in the pathogenesis of pruritic AD symptoms. However, the exact mechanism by which AMP-IBP5 regulates itch sensation in the context of AD, especially the role of LRP1, needs to be clarified.

In summary, we demonstrated that AMP-IBP5 enhanced TJ barrier function in human keratinocytes and ameliorated dermatitis symptoms and restored skin barrier function in a mouse model of AD through LRP1. These results suggest a potential therapeutic approach for AD using AMP-IBP5.

## 4. Materials and Methods

### 4.1. Reagents

AMP-IBP5 (AVYLPNCDRKGFYKRKQCKPSR-NH_2_; molecular weight: 2655.1) was obtained from the Peptide Institute (Osaka, Japan) and was dissolved in acetic acid (0.01%). IgG isotype control and antibodies specific for phosphorylated and unphosphorylated aPKCζ and Rac1 were purchased from Cell Signaling Technology (Beverly, MA, USA). A rabbit monoclonal antibody specific for claudin-1 was purchased from Cell Signaling Technology. Mouse monoclonal antibodies specific for claudin-4, occludin, and ZO-1 were obtained from Invitrogen (Carlsbad, CA, USA). A rabbit polyclonal antibody specific for claudin-7 was purchased from Invitrogen. The aPKCζ inhibitor GF 109203X was obtained from Enzo Life Sciences (Farmingdale, NY, USA). The Rac1 inhibitor NSC23766 was obtained from Calbiochem (La Jolla, CA, USA). Enzyme-linked immunosorbent assay (ELISA) kits were obtained from R&D Systems (Minneapolis, MN, USA). The EZ-link^TM^ Sulfo-NHS-LC-Biotin tracer was purchased from Thermo Scientific (Waltham, MA, USA). Recombinant human and mouse LRPAP were obtained from R&D Systems. Toluidine blue solution (0.05%, pH 4.1) was purchased from Muto Pure Chemicals Co. Ltd. (Tokyo, Japan). Recombinant human IL-4 and IL-13 were obtained from BioLegend (San Diego, CA, USA). A mouse monoclonal antibody specific for CD4 (PE rat anti-mouse CD4) was purchased from BD Bioscience (San Jose, CA, USA).

### 4.2. Keratinocyte Culture and Stimulation

Primary human epidermal keratinocytes were purchased from Kurabo Industries (Osaka, Japan) and were cultured in serum-free HuMedia-KG2 keratinocyte growth medium, as described in a previous study [72]. An increase in the Ca^2+^ concentration in cultured keratinocytes leads to the formation of TJs and enhancement of skin barrier function [32]. Thus, keratinocytes were cultured in high-Ca^2+^ (1.8 mM) medium to generate the TJ-forming keratinocytes of the second layer of the SG [73]. To establish the in vitro model with dysfunction of the TJ barrier, keratinocytes were stimulated by the addition of 100 ng/mL recombinant IL-4 and IL-13 in combination with the culture medium, as reported previously [74,75].

### 4.3. Animals

To induce AD-like inflammation in BALB/c mice, a strategy of cutaneous DNCB sensitization and challenge was used. Six-week-old female BALB/c mice were obtained from Japan SLC, Inc. (Tokyo, Japan) and housed under specific pathogen-free controlled conditions. DNBC was applied topically to the same area of skin on each mouse to induce AD-like skin lesions as described in a previous study [76]. In brief, 1% DNCB was applied to the ear skin of mice on Day 4. Four days later, the mice were challenged with 0.4% DNCB on the same area of skin 3 times weekly for 3 weeks (Days 1–19). The total dermatitis score (maximum score 12), representing the clinical severity, was defined as the sum of the individual scores (0 (none), 1 (mild), 2 (moderate), and 3 (severe)) assigned for each of 4 symptoms (erythema/hemorrhage, scaling/dryness, edema, and excoriation/erosion). To reduce variability, clinical symptoms were evaluated by two examiners blinded to the treatment conditions of the groups. The ear thickness of the mice was evaluated by measuring the entire thickness of the ear at the widest point from the outermost layer of one SC to the outermost layer of another SC on the opposite side of the ear using a micrometer (Mitutoyo, Kawasaki, Japan). TEWL was determined by a Tewameter TM Nano measurement probe (Courage + Khazaka electronic GmbH, Köln, Germany).

### 4.4. Treatment of Mice

The ear skin of the AD mice was subcutaneously injected with 25 μL of 25 μM AMP-IBP5 on Days 15, 17, and 18. In other experiments, AD mice were subcutaneously co-injected with AMP-IBP5 and 1 μg/mL recombinant mouse LRPAP on Days 15, 17, and 18. To compare the effects of injection and topical treatment, 25 μL of 25 μM AMP-IBP5 was topically applied to the ear skin of AD mice on Days 15, 17, and 18 (Appendix A). On Day 19, serum and skin biopsies were collected and analyzed.

### 4.5. Histological Analysis

Mouse ear tissues were fixed with 20% neutral buffered formalin solution, embedded in paraffin, and sectioned with a microtome. The slides were stained with hematoxylin and eosin (H&E) for histopathologic analysis. Mast cells were stained with 0.05% toluidine blue. Images were acquired using a Zeiss microscope (Carl Zeiss, Jena, Germany) and were analyzed with ImageJ software (version 1.52a, National Institutes of Health (NIH), Bethesda, MD, USA).

### 4.6. Western Blot Analysis

Samples of human keratinocytes and mouse skin tissues were harvested and were then lysed with RIPA lysis buffer (20 mM Tris-HCl (pH 7.5), 150 mM NaCl, 1 mM Na_2_ EDTA, 1 mM EGTA, 1% NP-40, 1% sodium deoxycholate, 2.5 mM sodium pyrophosphate, 1 mM β-glycerophosphate, 1 mM Na_3_VO_4_, and 1 μg/mL leupeptin). Protein concentrations were measured using Precision Red Advanced Protein Assay reagent, and equal amounts of total protein were separated by electrophoresis on 8–15% SDS–PAGE gels and then transferred to polyvinylidene fluoride membranes (Merck Millipore, Burlington, MA, USA). ImmunoBlock buffer was used to block the membranes for 1 h at room temperature. After blocking, the membranes were incubated at 4 °C with primary antibodies specific for claudin-1, -4, -7, occludin, and ZO-1. The primary antibodies were then detected using horseradish peroxidase-conjugated sheep anti-rabbit or sheep anti-mouse secondary antibodies. Membranes were developed with Luminata Forte Western horseradish peroxidase substrate (Merck Millipore, Burlington, MA, USA) and were then imaged using Fujifilm LAS-4000 Plus (Fujifilm, Tokyo, Japan). Band densities in the images were quantified using ImageJ.

### 4.7. Total RNA Extraction and Real-Time PCR

Total RNA was extracted from keratinocytes and skin tissues using the RNeasy Plus Micro Kit (QIAGEN, Hilden, Germany) and RNeasy Plus Universal Mini Kit (QIAGEN), respectively. Reverse transcription of 1 μg of total RNA to first-strand cDNA was performed using ReverTra Ace qPCR RT Master Mix (Toyobo, Osaka, Japan) or ReverTra Ace qPCR RT Master Mix with gDNA Remover (Toyobo) according to the manufacturer’s instructions. Real-time PCR was performed using the QuantiTect SYBR Green PCR Kit (QIAGEN). The StepOnePlus Real-Time PCR System (Life Technologies, Carlsbad, CA, USA) was used to quantify the mRNA levels in the samples. The primer sequences used in this study are listed in Table 1. All primers were obtained from Thermo Fisher Scientific (Waltham, MA, USA). The target RNA levels were normalized to the endogenous *Rps18* reference level, and changes in mRNA expression are reported as fold increases relative to vehicle.

### 4.8. Immunostaining Analysis

Keratinocytes were cultured on collagen I-coated chamber slides (BD Biosciences, Bedford, MA, USA) and subsequently fixed with methanol. Protein Block Serum-Free containing 0.2% Tween 20 was used for blocking prior to incubation overnight with the appropriate primary antibodies in 1% bovine serum albumin (BSA) and phosphate-buffered saline (PBS) with 0.2% Tween 20, followed by incubation with specific secondary antibodies conjugated to Alexa 594 (Invitrogen, Carlsbad, CA, USA). Confocal laser scanning microscopy (Carl Zeiss, Jena, Germany) was used for image acquisition.

Tissues harvested from mice were directly embedded in an optimal cutting temperature compound, and frozen sections were fixed with preheated 4% paraformaldehyde in PBS for 10 min. The sections were then permeabilized with 0.01% Triton X-100 in PBS for 10 min, blocked with ImmunoBlock for 30 min, and incubated overnight at 4 °C with the appropriate primary antibodies. After incubation with secondary antibodies, the samples were mounted using antifade mountant with NucBlue stain (Invitrogen, Carlsbad, CA, USA). Images were acquired using a Zeiss LSM 780 system with ZEN 2011 software (Carl Zeiss, Jena, Germany). ImageJ software was used to quantify the immunofluorescence intensities in the samples.

### 4.9. ELISA

To determine the level of total IgE in mouse serum, mouse serum was collected, and the total IgE level was measured as follows. Ninety-six-well plates were coated with 2 μg/mL purified rat anti-mouse IgE overnight at 4 °C and blocked with 20% ImmunoBlock at 37 °C for 90 min. The samples and purified mouse IgE used as the standard were added to assay wells and incubated at 37 °C for 80 min. After incubation with horseradish peroxidase-conjugated anti-mouse IgE, TMB substrate was added to the assay wells for 20 min. Then, stop solution containing 1 M sulfuric acid was added. Finally, the optical density of each assay well was measured at 450 nm.

### 4.10. Microarray Analysis

The microarray datasets GSE27887, GSE32924, GSE36842, GSE58558, GSE59294, GSE95759, GSE99802, GSE107361, GSE120899, GSE130588, GSE133385, GSE133477, and GSE140684 were obtained from the Gene Expression Omnibus database. The data were analyzed with Transcriptome Analysis Console software 4.0 (Applied Biosystems, Thermo Fisher Scientific, Waltham, MA, USA).

### 4.11. TJ Permeability Assay

The surface biotinylation technique was used to perform the TJ permeability assay, as described in a previous study [41]. In brief, EZ-link^TM^ Sulfo-NHS-LC-Biotin in PBS containing 1 M CaCl_2_ was injected dermally into the lesional or non-lesional skin of mice. After 30 min of incubation, the skin was harvested and immediately embedded in optimal cutting temperature compound. Frozen sections (5 μm) were fixed with 4% paraformaldehyde in PBS for 10 min and permeabilized with 0.01% Triton X-100 in PBS for 10 min. The samples were then blocked with ImmunoBlock for 30 min and incubated overnight at 4 °C with primary antibodies specific for claudin-1. After washing, the sections were incubated with a mixture of an Alexa Fluor 488-conjugated goat anti-mouse antibody and streptavidin Alexa Fluor 594-conjugated antibody for 1 h. The sections were visualized as described above using a Zeiss LSM 780 system.

### 4.12. TER Assay

For the TER assay, 3.6 × 10^5^ keratinocytes were cultured on 0.6 cm^2^ Transwell filters and were then transferred into 1.8 mM high-Ca^2+^ medium. Then, 10 μM AMP-IBP5 was added to both the apical and basal compartments in the absence or presence of various inhibitors. Forty-eight hours after stimulation, the TER values of the keratinocyte monolayers were determined using a CellZscope system (NanoAnalytics, Münster, Germany).

### 4.13. Statistical Analysis

GraphPad Prism 9 software (GraphPad Software Company, version 9.0.0, San Diego, CA) was used for all statistical analyses. Student’s *t*-test was utilized for comparisons between two groups, and one-way analysis of variance (ANOVA) with Tukey’s multiple comparisons test was utilized for comparisons among multiple groups. *p* < 0.05 was considered to indicate a statistically significant difference.

## Figures and Tables

**Figure 1 ijms-24-05200-f001:**
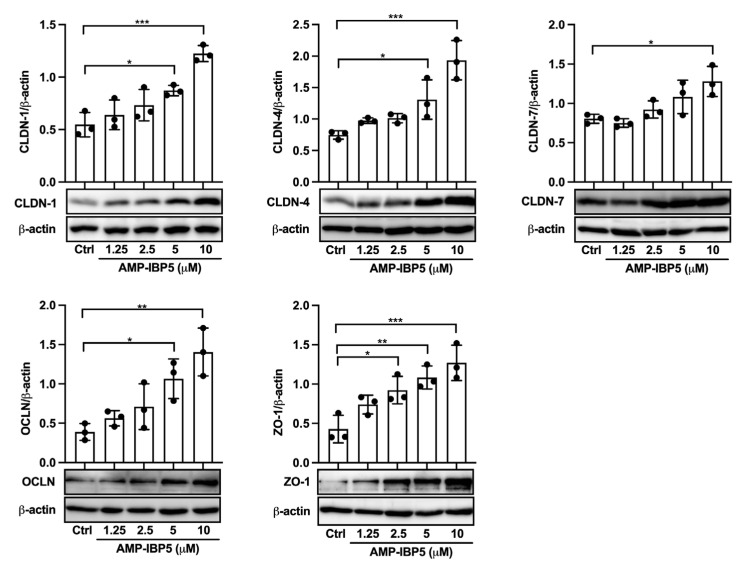
AMP-IBP5 increases the expression of TJ-related proteins. Human primary keratinocytes were incubated with 1.25, 2.5, 5, or 10 μM AMP-IBP5 for 48 h and subjected to Western blot analysis using antibodies specific for claudin-1, -4, -7, occludin, and ZO-1. Band densities were quantified using densitometry; *n* = 3/group. The data are presented as the means ± SDs. * *p* < 0.05, ** *p* < 0.01, *** *p* < 0.001. All data are representative of three independent experiments.

**Figure 2 ijms-24-05200-f002:**
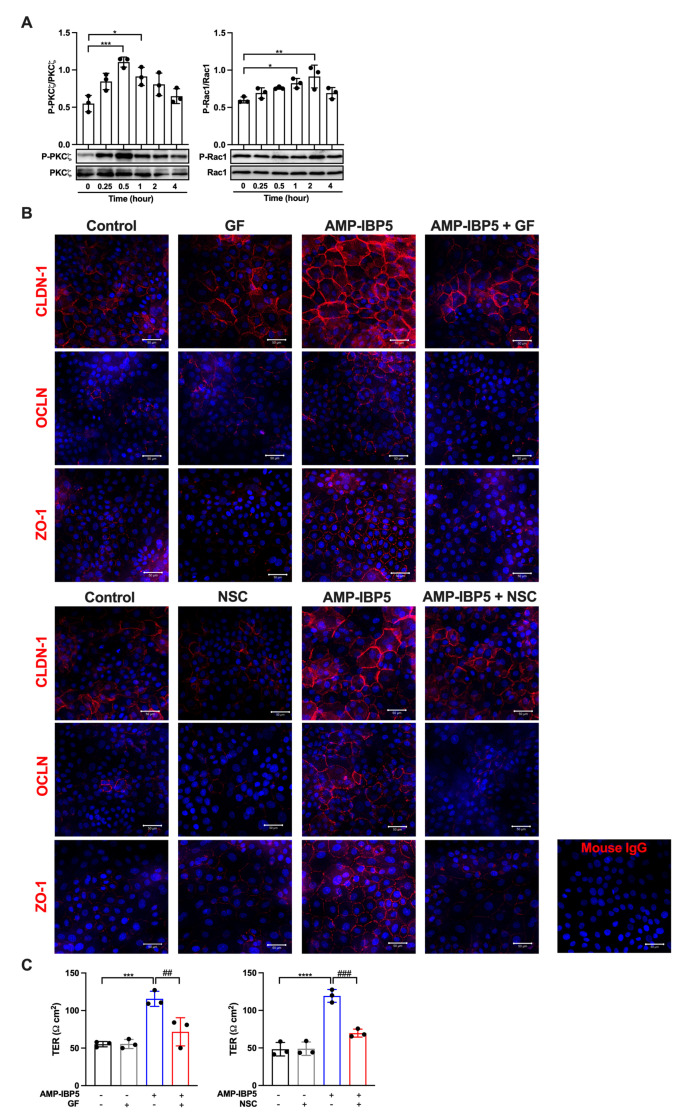
AMP-IBP5 regulates TJ barrier function in keratinocytes through activation of aPKCζ and Rac1. (**A**) Human primary keratinocytes were incubated with 10 μM AMP-IBP5 for 48 h and subjected to Western blot analysis using antibodies specific for phosphorylated and unphosphorylated aPKCζ and Rac1. Band densities were quantified using densitometry; *n* = 3/group. (**B**) Human primary keratinocytes were pretreated with GF 109203X (upper panel) or NSC23766 (lower panel) for 48 h and were then incubated with 10 μM AMP-IBP5 for 48 h. The results of one representative experiment of three independent experiments are shown. Scale bar: 50 μm. (**C**) Keratinocyte layers were pretreated with GF 109203X, NSC23766, or 0.1% DMSO for 48 h and were then incubated with 10 μM AMP-IBP5 for 48 h. Then, the TER was measured. The values obtained using stimulated and non-stimulated keratinocytes in the presence and absence of the inhibitor were compared. * *p* < 0.05, ** *p* < 0.01, *** *p* < 0.001, **** *p* < 0.0001, ^##^
*p* < 0.01, ^###^
*p* < 0.01.

**Figure 3 ijms-24-05200-f003:**
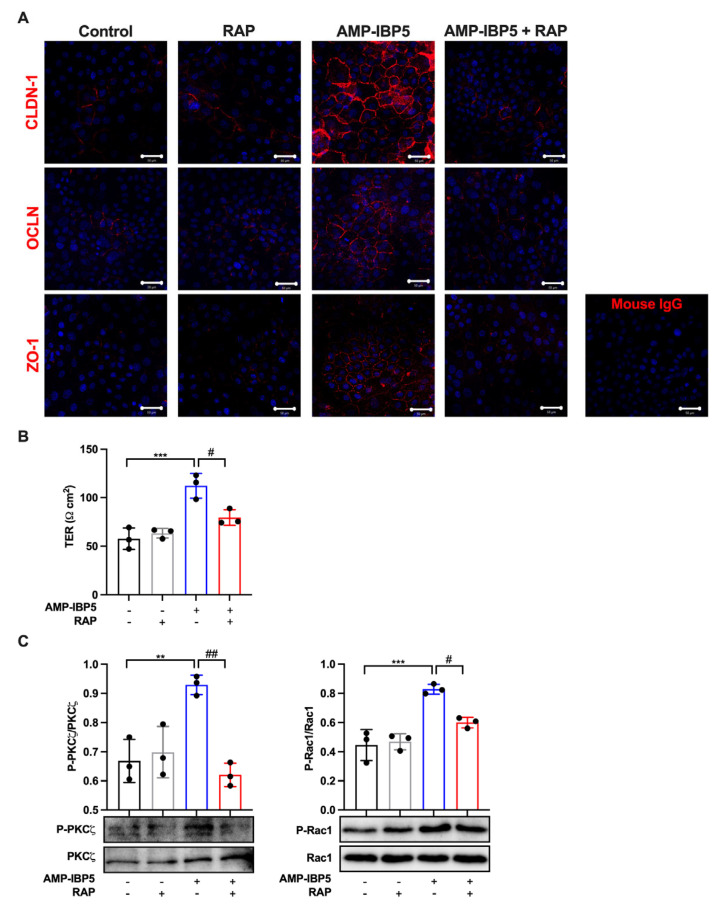
AMP-IBP5 regulates TJ barrier function in human keratinocytes via LRP1. Human primary keratinocytes were pretreated with 1 μg/mL recombinant human LRPAP protein (i.e., RAP) for 24 h and stimulated with 10 μM AMP-IBP5 for 48 h. (**A**) Immunofluorescence staining of the TJ-related proteins claudin-1, occludin, and ZO-1 upon stimulation with AMP-IBP5. The results of one representative experiment of three independent experiments are shown. Scale bar: 50 μm. (**B**) TER was measured after AMP-IBP5 treatment. The values obtained using stimulated and non-stimulated keratinocytes in the presence and absence of RAP were compared. (**C**) Immunoblot analysis using antibodies specific for phosphorylated and unphosphorylated aPKCζ and Rac1. ** *p* < 0.01, *** *p* < 0.001, ^#^ *p* < 0.05, ^##^ *p* < 0.01. All data are representative of three independent experiments.

**Figure 4 ijms-24-05200-f004:**
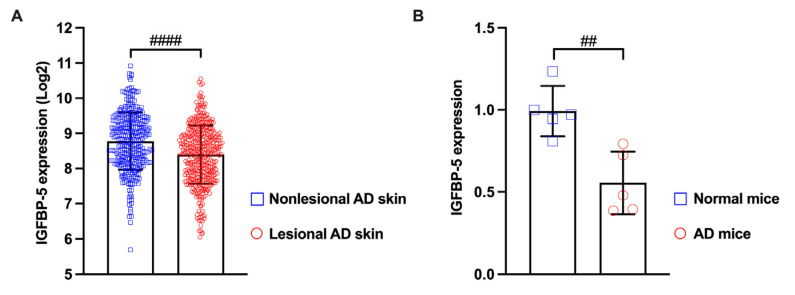
The expression of IGFBP-5 was decreased in lesional skin of patients with AD and in the AD mouse model. (**A**) The microarray datasets GSE27887, GSE32924, GSE36842, GSE58558, GSE59294, GSE95759, GSE99802, GSE107361, GSE120899, GSE130588, GSE133385, GSE133477, and GSE140684 were obtained from the Gene Expression Omnibus database. The samples included lesional skin (*n* = 413) and non-lesional skin (*n* = 344) of patients with AD. The data were analyzed with Transcriptome Analysis Console software 4.0 (Applied Biosystems, Thermo Fisher Scientific, Waltham, MA, USA). (**B**) The mRNA level of IGFBP-5 in AD mice was measured by real-time PCR. ^##^ *p* < 0.01, ^####^ *p* < 0.0001.

**Figure 5 ijms-24-05200-f005:**
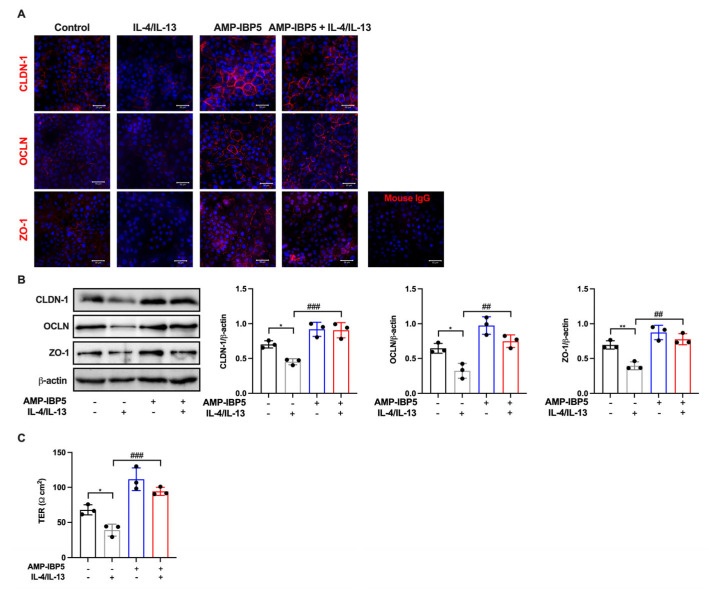
AMP-IBP5 rescues IL-4/IL-13-driven TJ barrier dysfunction. Human primary keratinocytes were pretreated with 100 ng/mL IL-4 and IL-13 in combination for 12 h and were then stimulated with 10 μM AMP-IBP5 for 48 h. (**A**) Immunofluorescence staining of the TJ-related proteins claudin-1, occludin, and ZO-1 upon stimulation with AMP-IBP5. The results of one representative experiment of three independent experiments are shown. Scale bar: 50 μm. (**B**) Immunoblot analysis using antibodies specific for claudin-1, occludin, and ZO-1. (**C**) TER was measured after AMP-IBP5 treatment. The values obtained using stimulated and non-stimulated cells in the presence and absence of IL-4/IL-13 were compared. * *p* < 0.05, ** *p* < 0.01, ^##^ *p* < 0.01, ^###^ *p* < 0.01. All data are representative of three independent experiments.

**Figure 6 ijms-24-05200-f006:**
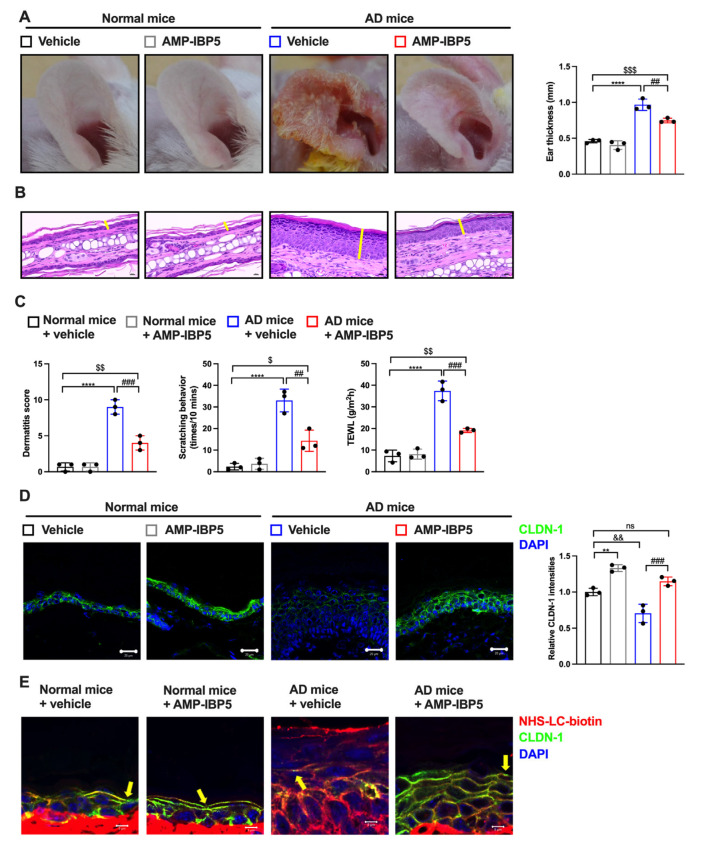
AMP-IBP5 ameliorates dermatitis-like symptoms and improves skin barrier function in AD mice. (**A**) Representative images of the ears from mice (left) and statistical analysis of ear thickness (right); n = 3 mice/group. (**B**) Representative histological sections of mouse ears stained with H&E. The yellow lines indicate the epidermis. Scale bars: 200 μm. (**C**) Dermatitis score, scratching behavior, and TEWL of mice. (**D**) Representative immunofluorescence images (left) and quantitative analysis of the claudin-1 staining intensity in the mouse epidermis (right). Scale bar: 20 μm; *n* = 3 mice/group. (**E**) A biotin tracer was also used to assess TJ permeability. Tracer exclusion is indicated by the yellow arrowheads. Scale bar: 5 μm; n = 3 mice/group. The data are presented as the means ± SDs. ** *p* < 0.01, **** *p* < 0.0001, ^##^ *p* < 0.01, ^###^ *p* < 0.001, ^$^
*p* < 0.05, ^$$^ *p* < 0.01, ^$$$^ *p* < 0.001, ^&&^ *p* < 0.01, ns: not significant. All data are representative of three independent experiments.

**Figure 7 ijms-24-05200-f007:**
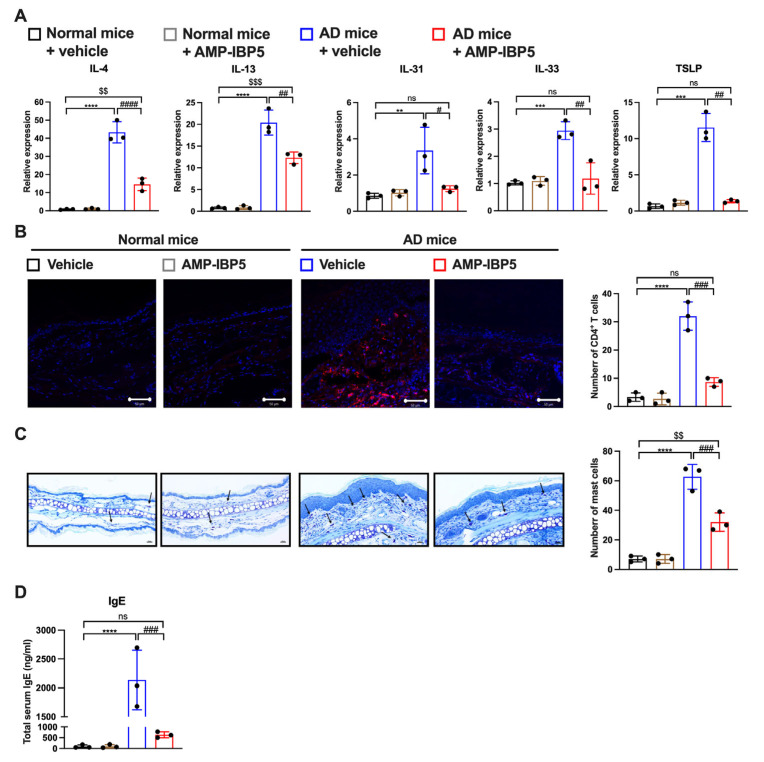
AMP-IBP5 treatment reduces the inflammatory reaction in AD mice. (**A**) The mRNA levels of IL-4, IL-13, IL-31, IL-33, and TSLP in AD mice were measured by real-time PCR. (**B**) Representative images of immunostaining of CD4^+^ T cells in AD mice (left) and the numbers of CD4^+^ T cells (right) in the stained sections. Scale bar: 50 μm; *n* = 3 mice/group. (**C**) Representative images of toluidine blue staining of mast cells (left) and the numbers of mast cells (right) in the stained sections. Scale bar: 100 μm; *n* = 3 mice/group. (**D**) The total serum IgE levels in AD mice were measured. The data are presented as the means ± SDs. ** *p* < 0.01, *** *p* < 0.001, **** *p* < 0.0001, ^#^
*p* < 0.05, ^##^ *p* < 0.01, ^###^ *p* < 0.001, ^####^ *p* < 0.0001, ^$$^ *p* < 0.01, ^$$$^ *p* < 0.001, ns: not significant. All data are representative of three independent experiments.

**Figure 8 ijms-24-05200-f008:**
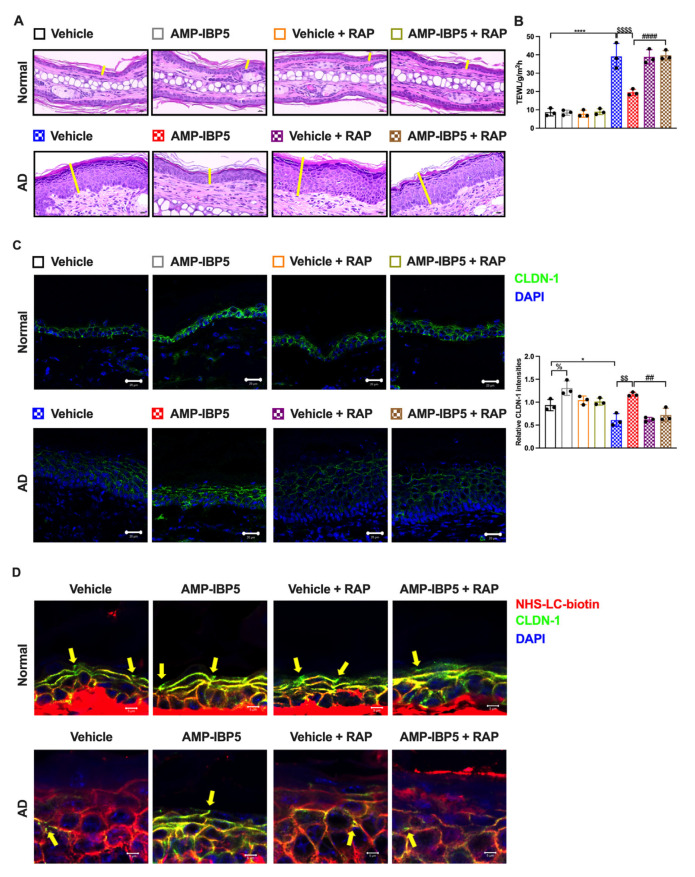
The restorative effect of AMP-IBP5 on skin barrier function is blocked by RAP injection in mice. (**A**) Representative images of H&E staining in AD mice co-injected with AMP-IBP5 and recombinant mouse LRPAP (i.e., RAP). Scale bar: 200 μm; *n* = 3/group. (**B**) TEWL in mouse ears on Day 19. (**C**) Representative immunofluorescence images (left) and quantitative analysis of the claudin-1 staining intensity in the mouse epidermis (right). Scale bar: 20 μm; *n* = 3 mice/group. (**D**) A biotin tracer was also used to assess TJ permeability. Tracer exclusion is indicated by the yellow arrowheads. Scale bar: 5 μm; *n* = 3 mice/group. The data are presented as the means ± SDs. * *p* < 0.05, **** *p* < 0.0001, ^##^ *p* < 0.01, ^####^ *p* < 0.0001, ^$$^ *p* < 0.01, ^$$$$^ *p* < 0.0001, ^%^ *p* < 0.05. All data are representative of three independent experiments.

**Figure 9 ijms-24-05200-f009:**
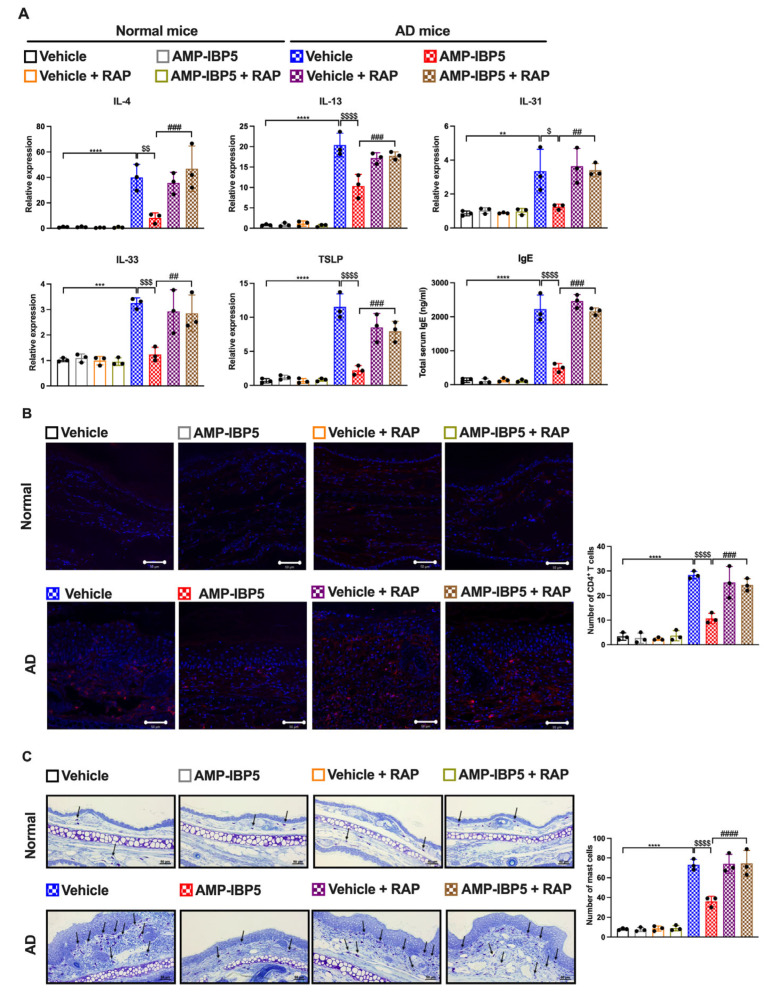
The suppressive effect of AM-IBP5 on inflammation is blocked by RAP injection in AD mice. (**A**) The mRNA levels of IL-4, IL-13, IL-31, IL-33, and TSLP and the total serum level of IgE in AD mice co-injected with AMP-IBP5 and recombinant mouse LRPAP (i.e., RAP) were measured. (**B**) Representative images of immunostaining of CD4^+^ T cells in AD mice (left) and the number of CD4^+^ T cells (right) in the stained sections. Scale bar: 50 μm; *n* = 3 mice/group. (**C**) Representative images of toluidine blue staining of mast cells, indicated by the black arrowheads (left) and the number of mast cells (right) in the stained sections. Scale bar: 50 μm; *n* = 3 mice/group. The data are presented as the means ± SDs. ** *p* < 0.01, *** *p* < 0.001, **** *p* < 0.0001, ^##^ *p* < 0.01, ^###^ *p* < 0.001, ^####^ *p* < 0.0001, ^$^ *p* < 0.05, ^$$^ *p* < 0.01, ^$$$^ *p* < 0.001, ^$$$$^ *p* < 0.0001. All data are representative of three independent experiments.

**Table 1 ijms-24-05200-t001:** Primers used for real-time PCR.

Gene Name	Primer Sequences
**Murine *Igfbp5***	Forward 5′-GTGTACCTGCCCAACTGTGACC-3′ Reverse 5′-GCAGCTTCATTCCGTACTTGTCC-3′
**Murine *Rps18***	Forward 5′-TTCTGGCCAACGGTCTAGACAAC-3′ Reverse 5′-CCAGTGGTCTTGGTGTGCTGA-3′
**Murine *IL-4***	Forward 5′-ACGGAGATGGATGTGCCAAAC-3′ Reverse 5′-AGCACCTTGGAAGCCCTACAGA-3′
**Murine *IL-13***	Forward 5′-CGGCAGCATGGTATGGAGTG-3′ Reverse 5′-ATTGCAATTGGAGATGTTGGTCAG-3′
**Murine *IL-31***	Forward 5′-TCCAGAGACCACAGGCAAAG-3′ Reverse 5′-AGGGTAGGCTTCGTTGTTCC-3′
**Murine *IL-33***	Forward 5′-GAGACTCCGTTCTGGCCTCA-3′ Reverse 5′-AATGTGTCAACAGACGCAGCAA-3′
**Murine *TSLP***	Forward 5′-CGAGCAAATCGAGGACTGTGAG-3′ Reverse 5′-GCAGTGGTCATTGAGGGCTTC-3′

## Data Availability

The data presented in this study are available on request from the corresponding author.

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
