# Peer review of "The Antimicrobial Peptide AMP-IBP5 Suppresses Dermatitis-like Lesions in a Mouse Model of Atopic Dermatitis through the Low-Density Lipoprotein Receptor-Related Protein-1 Receptor"

_ijms, 2023, doi:10.3390/ijms24065200_

Round 1

Reviewer 1 Report

The author reported that the antimicrobial peptide AMP-IBP5 might improve AD-like inflammation and enhance skin 29 barrier function through the low-density lipoprotein receptor-related 28 protein-1 (LRP1) receptor and suggested a possible implication of AMP-IBP5 in the treatment of AD. It is an exciting study; however, several questions must be straightforward first.

1.      What is the size of AMP-IBP5, and what is the vehicle?

2.      Section 4.4, which two groups of mice represented the subcutaneously injected AMP-IBP5 and topically applied the AMP-IBP5? The reviewer needs clarification about the results. Based on the 500 Dalton rule, the reviewer is unsure whether AMP-IBP5 could penetrate the skin barrier.

3.      For the in vivo study, why only select 25μM of AMP-IBP5 for treatment and only treat on Days 15, 17, and 18?

4.      Section 4.12, what is the dose of AMP-IBP5 for the TER study?

5.      The author performed the one-way analysis of variance (ANOVA) with Tukey’s multiple comparisons tests, which was utilized for comparisons of various groups but only showed partial statistical results in the figures. The reviewer is more interested in the difference between normal untreated mice and AD mice treated with AMP-IBP5. In addition, normal mice treated with AMP-IBP5 could be served as the safety study.

6.      The similarly rate is about 28% after excluding abstract, methods and reference, the author needs to rewrite the manuscript.

Author Response

Responses to Reviewer 1

General comment: The author reported that the antimicrobial peptide AMP-IBP5 might improve AD-like inflammation and enhance skin barrier function through the low-density lipoprotein receptor-related protein-1 (LRP1) receptor and suggested a possible implication of AMP-IBP5 in the treatment of AD. It is an exciting study; however, several questions must be straightforward first.

Response: We would like to thank Reviewer 1 for taking the time to read this manuscript, and for your kind appreciation on our work. Below, we provide point-by-point responses to the reviewer’s comments.

Comment 1: What is the size of AMP-IBP5, and what is the vehicle?

Response 1: The molecular weight of AMP-IBP5 is 2655.1 Dalton and the vehicle used to dissolve AMP-IBP5 is 0.01% acetic acid. This is found in Page 17, line 359 of the revised manuscript.

Comment 2: Section 4.4, which two groups of mice represented the subcutaneously injected AMP-IBP5 and topically applied the AMP-IBP5? The reviewer needs clarification about the results. Based on the 500 Dalton rule, the reviewer is unsure whether AMP-IBP5 could penetrate the skin barrier.

Response 2: We would like to thank Reviewer 1 for this comment and apologize for the poor explanation in the section 4.4. In this study, AMP-IBP5 was injected subcutaneously to the mouse skin. We performed 3 separate experiments to compare both the injection and topical treatment of AMP-IBP5, and the result is shown in Figure S5. We have added more details regarding the treatment of mice in our study as follows: “The ear skin of the AD mice was subcutaneously injected with 25 ml of 25 mM AMP-IBP5 on Days 15, 17, and 18. In other experiments, AD mice were subcutaneously coinjected with AMP-IBP5 and 1 mg/ml recombinant mouse LRPAP on Days 15, 17, and 18. To compare the effects of injection an topical treatment, 25 ml of 25 mM AMP-IBP5 was topically applied to the ear skin of AD mice on Days 15, 17, and 18 (Figure S5). On Day 19, serum and skin biopsies were collected and analyzed”.

Concerning how a 2655-Dalton AMP-IBP5 could penetrate the skin barrier, based on 500 Dalton rule, further studies are necessary to clarify this puzzle. However, we speculate that the fact that AMP-IBP5 used in this study was dissolved in acetic acid may have enhanced its penetration into the skin barrier. In fact, it has been demonstrated that acetic acid is a skin permeation enhancer (doi: 10.1016/j.jconrel.2003.10.016.; doi: 10.1159/000107575; doi: 10.17795/ajmb-18611; doi: 10.13040/ijpsr.0975-8232.5(8).3175-82;).

Comment 3: For the in vivo study, why only select 25μM of AMP-IBP5 for treatment and only treat on Days 15, 17, and 18?

Response 3: In the preliminary in vivo first experiments, we investigated the effect of various doses of AMP-IBP5 on DNCB-induced dermatitis-like mice and found that 25 μM was the lowest and effective concentration of AMP-IBP5 on AD mice to relieve the dermatitis-like symptoms. As for the treatment of AD mice on Days 15, 17, 18, this was based on the observations published in our previous study (doi:10.1007/s10753-017-0673-7). Days 15, 17 and 18 were a good timing to study the therapeutic effects of AMP-IBP5

Comment 4: Section 4.12, what is the dose of AMP-IBP5 for the TER study?

Response 4: We would like to apologize for the poor explanation in the section 4.12. We have added the dose of AMP-IBP5 for the TER study as follows: “Then, 10 mM AMP-IBP5 was added to both the apical and basal compartments in the absence or presence of various inhibitors”. This is cited in Page 21, line 491.

Comment 5: The author performed the one-way analysis of variance (ANOVA) with Tukey’s multiple comparisons tests, which was utilized for comparisons of various groups but only showed partial statistical results in the figures. The reviewer is more interested in the difference between normal untreated mice and AD mice treated with AMP-IBP5. In addition, normal mice treated with AMP-IBP5 could be served as the safety study.

Response 5: We would like to apologize for not showing all statistical results in the original version. In this revised manuscript, we have modified Figures 6 and 7 in which we added more statistical results between normal mice group and AD mice treated with AMP-IBP5 group. The results show that “Although significant differences remained between normal mice and AMP-IBP5-treated AD mice, these features of AD were markedly improved following AMP-IBP5 treatment”. This is found in Pages 9, lines 195-197.

Comment 6: The similarly rate is about 28% after excluding abstract, methods and reference, the author needs to rewrite the manuscript.

Response 6: We would like to thank Reviewer 1 for taking time and effort to read our manuscript. We have checked the manuscript carefully and rewrote all sentences needeed to decrease the similarity. The revised manuscript was also sent to a professional English editing service before re-submission (we have attached an Editing Certificate).

Reviewer 2 Report

The study reported on various aspects of AMP-IBP5 in alleviating the symptoms of atopic dermatitis specifically targeted on LRP1. The study is very well-designed and extensive. The results were reported in very detail and complete. Results were very well elaborated and easy to comprehend. 

Suggestion:

For Section 4.7, the presentation will be better if the primer sequence can be listed in a table form.

Author Response

Response to Reviewer 2

General comment: The study reported on various aspects of AMP-IBP5 in alleviating the symptoms of atopic dermatitis specifically targeted on LRP1. The study is very well-designed and extensive. The results were reported in very detail and complete. Results were very well elaborated and easy to comprehend. 

Response: We would like to thank Reviewer 2 for taking the time and effort necessary to review this manuscript. We sincerely appreciate your kind comments and suggestions. Below, we provide point-by-point responses to the reviewer’s comments.

Comment 1: For Section 4.7, the presentation will be better if the primer sequence can be listed in a table form.

Response 1: We totally agree with Reviewer’s comment. In the revised version, we have added a table which lists all primers sequences used in this study (Table 1. Primers used for real-time PCR). This is found in Page 19, line 443.

Reviewer 3 Report

Line 91: you are saying AMP-IBP5 improves the formation and function of the TJ barrier. You used normal human keratinocytes as the baseline setting. Why can you say "improve" the formation and function? Should you say -enhance, increase, elevate, etc?

Line 120: same as above. You should not use "improve".

Line 292: add "These experiments" on the immunopathogenesis of AD.... because there are many AD studies focusing on immune functions rather than barrier functions. You should focus on the study you conducted and described in the paper.

Line 293: Use "play" a crucial role rather than "show" a crucial role.

Line 338-9: I do not think you can definitively say the role of the LRP1 receptor in the pathogenesis of AD. LRP1 receptor involvement was suggested. Rewrite these two sentences.

Line 358 - : the discussion is lengthy. You did not show experiments using microbiomes. Shorten those discussions. The discussion should focus on the results you obtained from the studies since this paper covered many experiments, results, and figures. 

Author Response

Specific Responses:

Responses to Reviewer 3

We would like to thank Reviewer 3 for taking the time to read this manuscript. Below, we provide point-by-point responses to the reviewer’s comments.

Comment 1: Line 91: you are saying AMP-IBP5 improves the formation and function of the TJ barrier. You used normal human keratinocytes as the baseline setting. Why can you say "improve" the formation and function? Should you say -enhance, increase, elevate, etc?

Response 1: We totally agree with the Reviewer’s comment. In the revised version, we have changed “improves” to “enhances” (please see Page 3, line 92).

Comment 2: Line 120: same as above. You should not use "improve".

Response 2: As suggested, we have changed “improvement” to “enhancement” (Page 3, line 120) in the revised version of the manuscript.

Comment 3: Line 292: add "These experiments" on the immunopathogenesis of AD.... because there are many AD studies focusing on immune functions rather than barrier functions. You should focus on the study you conducted and described in the paper.

Response 3: Following Reviewer’s recommendation, we have changed the sentence as follows: “Skin barrier dysfunction contributes significantly to the pathogenesis of AD” (Page 15, line 275).

Comment 4: Line 293: Use “play” a crucial role rather than “show” a crucial role.

Response 4: In this revised version, we have changed “show” to “play” (Page 15, line 276).

Comment 5: Line 338-9: I do not think you can definitively say the role of the LRP1 receptor in the pathogenesis of AD. LRP1 receptor involvement was suggested. Rewrite these two sentences.

Response 5: We totally agree with the comment. In the revised version, we have changed the sentence as follows: “To our knowledge, this is the first report about the involvement of the LRP1 in the pathogenesis of AD. Further studies are required to investigate the mechanism by which LRP1 is involved in the pathogenesis of AD” (page 16, lines 320-322).

Comment 6: Line 358 - : the discussion is lengthy. You did not show experiments using microbiomes. Shorten those discussions. The discussion should focus on the results you obtained from the studies since this paper covered many experiments, results, and figures. 

Response 6: According to Reviewer’s comment, in the revised manuscript, we have deleted the paragraph discussing the possible effect of AMP-IBP5 on the microbiome in the treatment of AD.

Round 2

Reviewer 1 Report

The author has made sufficient improvement after 1st round of review; however, the reviewer suggested that the author should also show statistical results in figure s5. Interestingly, an over 500 Dalton compound penetrated through the skin barrier. Although the author points out it might be the enhanced effect of the penetration enhancer, it might also be the effect of the dysfunction of the skin barrier. A future diffusion cell study might be arranged to clarify this issue. 

Author Response

Specific Responses:

Responses to Reviewer 1

General comment: The author has made sufficient improvement after 1st round of review; however, the reviewer suggested that the author should also show statistical results in figure s5. Interestingly, an over 500 Dalton compound penetrated through the skin barrier. Although the author points out it might be the enhanced effect of the penetration enhancer, it might also be the effect of the dysfunction of the skin barrier. A future diffusion cell study might be arranged to clarify this issue. 

Response: We would like to thank Reviewer 1 for taking the time to read this manuscript, and for your kind appreciation on our work.

Following Reviewer’s recommendation, we have provided more statistical results in Figure S5 in the supplementary materials. In the injection group, we compared vehicle-treated normal mice versus vehicle-treated AD mice, vehicle-treated AD mice versus AMP-IBP5-treated AD mice, and vehicle-treated normal mice versus AMP-IBP5-treated AD mice. In the topical application group, we compared vehicle-treated normal mice versus vehicle-treated AD mice, vehicle-treated AD mice versus AMP-IBP5-treated AD mice, and vehicle-treated normal mice versus AMP-IBP5-treated AD mice. This is found in the figure legend of Figure S5, in Page 6, lines 54-63 of the supplementary materials.

In addition, we totally agree with the Reviewer that AMP-IBP5 may easily penetrate into the skin because of dysfunctional skin barrier. In the discussion of the re-revised version, we described this issue as follows: “There was no significant difference in alleviation of dermatitis-like symptoms between subcutaneous injection and topical application of AMP-IBP5. According to 500 Dalton rule for the skin penetration of chemical compounds, AMP-IBP5 with a molecular weight of 2655 Dalton hardly penetrates the skin. We speculated that AMP-IBP5 succeeded in penetrating the skin because it was dissolved in acetic acid, which is known as a skin penetration enhancer. Moreover, skin barrier impairment in AD may also facilitate the penetration of AMP-IBP5 into the skin. Further studies are required to clarify the topical effect of AMP-IBP5 in AD pathogenesis”. This is found in Page 15, lines 297-304 of the revised manuscript (in red). Corresponding references are 47, 48, 49.

We hope that these re-revisions are satisfactory and that the manuscript is suitable for publication in the IJMS.

Sincerely,

François Niyonsaba, M.D., Ph.D.

Atopy (Allergy) Research Center

Juntendo University School of Medicine

2-1-1 Hongo, Bunkyo-ku, Tokyo 113-8421, Japan

Tel.: +81-3-5802-1591; Fax: +81-3-3813-5512

E-mail address: francois@ juntendo.ac.jp
